# Dual-Hormone Insulin-and-Pramlintide Artificial Pancreas for Type 1 Diabetes: A Systematic Review

**Alexandra Torres-Castaño** [1,2,3,4,*], **Amado Rivero-Santana** [1,2,3,4], **Lilisbeth Perestelo-Pérez** [1,2,3,4], **Andrea Duarte-Díaz** [1,2,3,4], **Analia Abt-Sacks** [1,2,3,4], **Vanesa Ramos-García** [1,2,3,4], **Yolanda Álvarez-Pérez** [1,2,3,4], **Ana M. Wäagner** [5], **Mercedes Rigla** [6] and **Pedro Serrano-Aguilar** [2,3,4]

1. Canary Islands Health Research Institute Foundation (FIISC), 38109 El Rosario, Spain
2. Evaluation Unit of the Canary Islands Health Service (SESCS), 38019 El Rosario, Spain
3. The Spanish Network of Agencies for Health Technology Assessment and Services of the National Health System (RedETS), 28071 Madrid, Spain
4. Network for Research on Chronicity, Primary Care and Health Promotion (RICAPPS), 28029 Madrid, Spain
5. Facultad de Ciencias de la Salud, Universidad de Las Palmas de Gran Canaria, 35017 Las Palmas de Gran Canaria, Spain
6. Consorci Hospitalari Parc Taulí, 08208 Sabadell, Spain
* Correspondence: atorrcas@sescs.es

**Abstract:** The artificial pancreas (AP) is equipped with a glucose monitoring sensor, an insulin pump and an integrated mathematical algorithm that determines insulin infusion based on the glucose levels detected by the sensor. Research has shown that AP can help patients with type-1 Diabetes Mellitus (T1DM) to improve the control of their glucose levels, but the occurrence of postprandial hyperglycemia is still considerable. The addition of pramlintide (a synthetic derivative analog of amylin) in a dual-hormone AP could improve postprandial glycemic control. This systematic review aims to evaluate and synthesize the evidence on the safety, efficacy and cost-effectiveness of the dual insulin- and pramlintide-releasing AP. The electronic databases MEDLINE, Embase, Web of Science and ClinicalTrials.gov were consulted up to 6 June 2021. We identified four small crossover studies (n = 59) and two ongoing crossover trials, all of them carried out by the same research group. The four studies observed more gastrointestinal adverse effects with the dual system. One study found that the dual system improved outcomes compared to insulin alone, with precise carbohydrate counting (CC) in both groups. Another study showed that a fully closed-loop system (without CC) was equivalent to an insulin-alone AP (with CC) on time in the target range but performed worse in hyperglycemia during the daytime. These preliminary results suggest that the control of postprandial hyperglycemia remains a challenge.

**Keywords:** artificial pancreas; dual release; glycemic control pramlintide

## 1. Introduction

Successful management of type 1 Diabetes Mellitus (T1DM) requires that patients learn to plan daily activities such as physical exercise and diet and to measure their glucose levels regularly in order to avoid acute and long-term complications, which produce a high burden on patients' quality of life [1]. Monitoring devices and insulin pumps developed in the last decades with the functionality to continuously measure interstitial glucose and adjust insulin infusion are helpful in improving glucose control clinical outcomes [2,3].

The artificial pancreas (AP) represents the most recent evolution of these devices, aimed at simplifying and improving care for patients with T1DM. The AP combines a glucose monitoring sensor (which is attached to the arm or abdomen and measures interstitial glucose concentrations), a transmitter that sends these data to an insulin pump and an integrated mathematical algorithm mounted on smartphones or tablets that processes and

models continuous glucose monitoring (CGM) data to provide the appropriate insulin infusion based on the glucose levels detected by the sensor. In addition to suspending insulin delivery when glucose levels fall below a predetermined threshold, as in previous devices that integrated CGM and insulin pumps (i.e., sensor-augmented pump) [2,4], the AP also increases insulin dosing when hyperglycemic levels are reached to mimic the functioning of the biological pancreas [5]. Single-hormone versions only deliver insulin, whereas dual-hormone versions also include the infusion of glucagon (requiring an additional catheter and infusion pump) [6–9].

Research has shown that AP increases time in the glucose target range and reduces time in hypo- and hyperglycemic levels compared to insulin pump treatment or sensor-augmented pumps [10–12], although the quality of this evidence is limited due to the short follow-up of studies carried out to date. Compared to single-hormone AP, dual-hormone systems have shown a slightly lower time in hypoglycemia, but no differences in the time in the target range and more gastrointestinal symptoms [13]

Most versions of currently developed APs are not fully automated closed-loop systems; they still require the patient to provide information on physical activities and meals in order to adjust the insulin infusion (hybrid systems). Despite the benefits mentioned above, studies show that significant durations of hyperglycemia (5–8 h/day above 10 mmol/L [180 mg/dL]) are still reported, particularly after meals [14,15]. Therefore, manual adjustments and carbohydrate counting (CC) are still required to calculate insulin doses for each meal. The need for accurate CC can be a barrier for patients, making them feel restrained or anxious or influencing dietary choices in favor or pre-packaged processed foods [16]. An alternative to CC in the use of AP is the "simple meal announcement" (SMA), whereby the system delivers partial boluses of insulin regardless of the carbohydrate content to be ingested. However, this has resulted in higher levels of postprandial hyperglycemia compared to CC [17].

Pramlintide is a synthetic derivative analog of amylin, a hormone released by the beta cells of the pancreas that is co-secreted with insulin after a meal in healthy individuals but is deficient in people with T1DM [8]. Three effects of this hormone are highly relevant to the treatment of diabetes: (1) modulation of gastric emptying, which can be abnormally rapid in diabetes; (2) suppression of glucagon, which is excessively secreted in diabetes, especially after meals; (3) development of satiety shortly after starting to eat. Together with insulin, these effects limit post-meal hyperglycemia and prevent calorie intake [18]. Pramlintide is injected with meals because it lowers postprandial glycemia, allowing patients to reduce insulin doses [19–21].

The use of an AP with dual infusion of insulin and pramlintide could represent a helpful resource in the control of postprandial blood glucose levels in complex diabetic patients. This systematic review, therefore, aims to explore the available evidence about the effectiveness and safety of the use of an AP with dual release of insulin and pramlintide for the treatment of people with type 1 diabetes.

## 2. Materials and Methods

A systematic review of the literature was conducted in accordance with the Preferred Reporting Items for Systematic Reviews and Meta-Analysis (PRISMA) statement [22]. The detail of the PRISMA checklist can be found in Supplementary Table S1. The systematic review was prospectively registered in the International Prospective Register of Systematic Reviews (PROSPERO) with registration number CRD42022290673.

### 2.1. Search Strategy

First, a preliminary manual search was carried out to locate possible health technology assessment (HTA) reports and/or previous systematic reviews on the subject that could provide background information. Secondly, Medline (Ovid SP), Embase (Elsevier) and Web of Science (Clarivate Analytics) were searched for potentially eligible articles published up to 6 June 2021. An overall search strategy was developed using subject headings and

free text terms and then adapted for each database to ensure sensitivity. As an example, the MEDLINE search strategy is shown in Table 1. Search strategies for the other two electronic databases are available in Supplementary Table S2. No language or publication year restrictions were applied to limit the search strategy. In addition, a manual search was performed at the ClinicalTrials.gov website in October 2021 for a complete identification of ongoing studies. Details are also available in Supplementary Table S3.

**Table 1.** Medline search strategy.

| 1 | Diabetes Mellitus, Type 1/ | 78,504 |
|---|---|---|
| 2 | exp Diabetic Ketoacidosis/ | 6745 |
| 3 | (diabet\$ adj3 (britt\$ or juvenil\$ or pediatric or pediatric or child\$ or early or keto\$ or labil\$ or acidos\$ or autoimmun\$ or auto immun\$ or sudden onset or typ\$ 1 or typ\$ I)).ti,ab,hw. | 116,952 |
| 4 | (insulin depend\$ or insulindepend\$ or insulin-depend\$).ti,ab,kw. | 29,716 |
| 5 | (IDDM or T1DM or T1D or dm1 or dm 1 or dmt1 or dm t1 or t1 dm).ti,ab,kw. | 24,116 |
| 6 | 1 or 2 or 3 or 4 or 5 | 136,479 |
| 7 | exp Diabetes Insipidus/ | 8025 |
| 8 | diabet\$ insipidus.tw. | 8905 |
| 9 | 7 or 8 | 11174 |
| 10 | 6 not 9 | 135,897 |
| 11 | Pancreas, Artificial/ | 842 |
| 12 | artificial pancreas.ti,ab. | 1292 |
| 13 | 11 or 12 | 1626 |
| 14 | pramlintide.ti,ab. | 359 |
| 15 | 13 and 14 | 10 |
| 16 | ((single or dual) adj3 hormon*).ti,ab. | 1122 |
| 17 | 13 and 16 | 54 |

### 2.2. Inclusion and Exclusion Criteria

Studies were included if they met the following criteria: (1) participants of all ages with Type 1 Diabetes Mellitus, (2) treated with dual-hormone insulin-and-pramlintide AP; (3) insulin-alone AP as control; and (4) designed as systematic reviews, randomized controlled trials (RCT), non-randomized controlled trials (NRCT) or observational studies (both prospective and retrospective). Studies with the following characteristics were excluded: (1) disease other than type 1 diabetes, (2) diabetes management devices other than AP, and APs that did not have dual release of insulin and pramlintide. (3) other outcome measures not related to the effectiveness and safety of the AP, (4) designed as narrative reviews, editorials, letters to the editor, opinions, qualitative studies, or conference abstracts.

### 2.3. Study Selection

The citations retrieved from the electronic databases were imported into a standardized Microsoft Excel data sheet and duplicates were removed. First, all titles and abstracts were screened in order to pre-select those meeting the inclusion criteria. Full texts for all the potentially relevant articles were retrieved. Then, the full texts of these studies were analyzed in depth. The study selection process was conducted independently by two authors, and any disagreement was solved through discussion and consensus or through consultations with a third reviewer if disagreement persisted. The bibliographic references were stored using the Reference Manager Version 10® (Thomson Scientific, Philadelphia, PA, USA).

### 2.4. Data Extraction and Quality Assessment

The following items were extracted from all included studies using a pre-specified data extraction form in Microsoft Excel: first author, year of publication, participants' characteristics, intervention and comparator details, outcome measures and conflict of interest. Data extraction was performed by one reviewer and checked by another, and the possible discrepancies were resolved through discussion. The risk of bias in the included studies was assessed using version 2 of the Cochrane risk-of-bias tool for randomized trials (ROB-2) [23]. Quality assessment was undertaken by two independent reviewers, and disagreements were solved by discussion and consensus or after consulting a third reviewer.

### 2.5. Data Synthesis

A narrative synthesis of the results of each individual study was conducted.

### 3. Results

In total, 270 publications were identified from the literature search, including 57 from Medline, 112 from EMBASE, and 101 from Web of Science; 101 of these were duplicates and were removed. Hence, 169 unique articles were identified, and after reviewing titles and abstracts, 18 were selected for full-text review. Three of these articles met the inclusion criteria and were finally included in this review. Figure 1 shows the PRISMA flowchart of the study selection process.

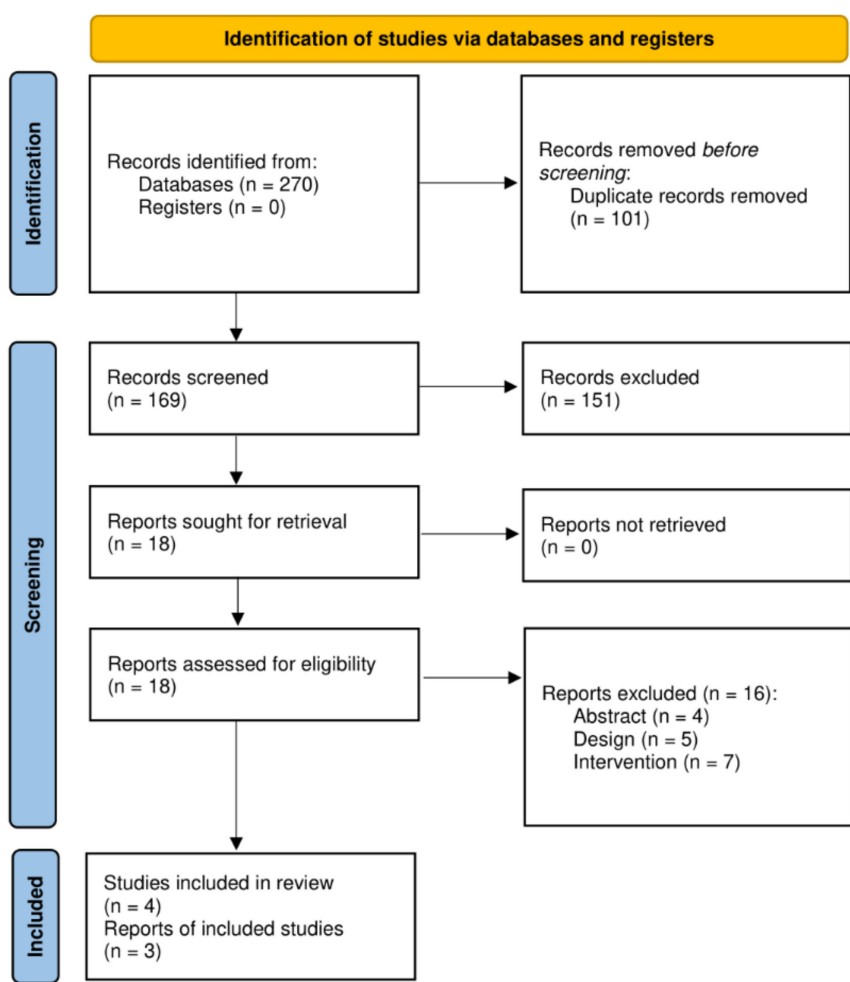

**Figure 1.** PRISMA flowchart of the study selection process.

### 3.1. Study Characteristics

The characteristics of the four included studies are shown in Table 2. All of them were crossover RCTs conducted by the same research group. Haidar et al. (2020) [8] (n = 24) compared an AP with rapid insulin and pramlintide, AP with regular insulin and pramlintide, and AP with rapid insulin alone (in all cases with CC) over one hospital day. Tsoukas et al. (2021) [24] included two small studies. The first was a feasibility study (n = 7) comparing a dual system with Faster acting insulin Aspart (FiASP) and pramlintide (with SMA) versus FiASP alone (with CC), also over one hospital day. The second was a pilot study (n = 7) that compared the same interventions as in the feasibility study over three 12-day phases. However, the FiASP alone phase also included a placebo, and there was a third phase of FiASP with placebo and SMA. Finally, Tsoukas et al. (2021b) [25] (n = 28) compared FiASP and pramlintide during one day of hospitalization with FiASP alone with CC.

All four studies included a washout period between interventions of 14–45 days. In the study by Haidar et al. (2020) [8], pramlintide was administered at a fixed rate of 6 μg/insulin unit, whereas 10 μg/insulin unit was used in the remaining studies (since no CC was performed in these studies, the bolus was smaller, and therefore, the ratio was increased to administer an amount similar to pramlintide at meals). Pramlintide was administered by means of an independent infusion pump to that of insulin.

**Table 2.** Characteristics of the included studies.

| Author (Year), Country | Study Design | Population | Intervention | Duration | Comparator | Outcome | Conflict of Interest |
|---|---|---|---|---|---|---|---|
| Haidar (2020) [8], Canada | RCT (crossover) | N = 24 T1DM | AP with rapid insulin-and-pramlintide | Three 24-h inpatient visits Each visit was preceded by an outpatient hormonal open-loop run-in period of 10–14 d | AP with regular insulin-and-pramlintide AP with rapid insulin alone | Time in target range (70–180 mg/dL) Time in hypoglycemia Time in hyperglycemia Mean glucose level (mg/dL) Glucose variability Insulin units | The authors received research support/consulting fees from different pharmaceutical industries. |
| Tsoukas (2021) [25], Canada | RCT (crossover) | N = 24 T1DM | AP with FiASP-and-pramlintide | Two 27-h inpatient visits | AP with FiASP alone | Time in target range (70–180 mg/dL) Time in hypoglycemia Time in hyperglycemia Mean glucose level (mg/dL) Glucose variability Insulin units | The authors received research support/consulting fees from medical industries (Lily, Eli, Adocia, Aga Matrix, Novo Nordisk, Boeringher Ingelheim, Janssen and AstraZeneca) and insulin pumps, glucose sensors, and monitors from Dexcom, Tandem, and Medtronic. |
| Tsoukas (2021) [24], Canada | Feasibility study | N = 7 (4 adults and 3 adolescents) T1DM | AP with FiASP-and-pramlintide + SMA | 24h inpatient visits | AP with FiASP + FCC | Gastrointestinal symptoms | The authors received research support/consulting fees from different pharmaceutical industries (Eli Lilly, Novo Nordisk, Boehringer Ingelheim, Janssen and AstraZeneca) and received consulting fees from Dexcom and Insulet. One author has pending patents in the artificial pancreas area. |
| Tsoukas (2021) [24], Canada | Pilot study | N = 4 T1DM | AP with FiASP-and-pramlintide + SMA | 12 d | AP with FiASP-and-placebo + FCC AP with FiASP-and-placebo + SMA | Time in target range (70–180 mg/dL) Time in hypoglycemia Time in hyperglycemia Mean glucose level (mg/dL) Glucose variability Insulin units DDS HFS-w INSPIRE DBSQ | |

AP = artificial pancreas; DBSQ = the diabetes bowel symptom questionnaire; DDS = diabetes distress scale; FCC = full carbohydrate counting; FiASP = fast-acting insulin aspart; HFS-w = hypoglycemia fear survey—worry subscale; RCT = randomized controlled trial; SMA = simple meal announcement; h = hours; d = days; T1DM = Type 1 Diabetes Mellitus.

### 3.2. Quality Assessment

An assessment of the risk of bias of these studies was undertaken using the Cochrane Risk of Bias (RoB-2) tool [23], which is shown in detail in Supplementary Table S4. Overall, four studies were rated at unclear risk of bias. The reason for this is related to lack of blinding, both for participants and for those administering the intervention, and lack of information on protocol deviations. Furthermore, missing data exceeded 5% of the total. For the feasibility study by Tsoukas et al. [24], the main concerns were a lack of information about the randomization procedure and the number of participants in each intervention. Figure 2 shows the risk of bias summary. Figure 3 shows each reviewer's assessment in relation to each item presented as percentages across all included studies.

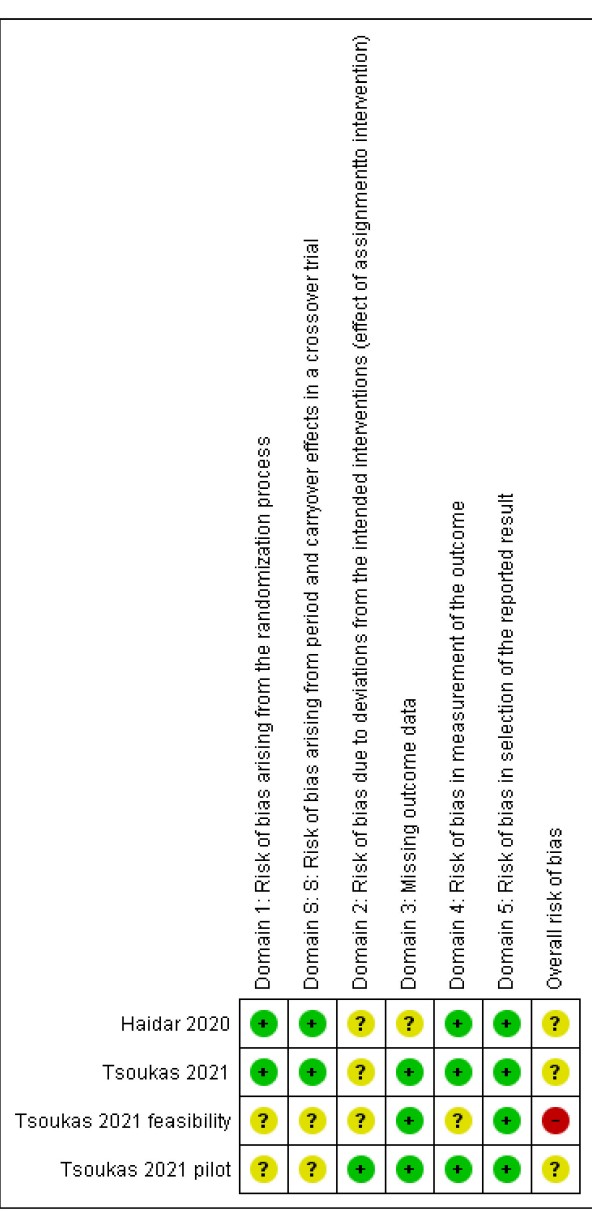

**Figure 2.** Risk of bias summary [8,24].

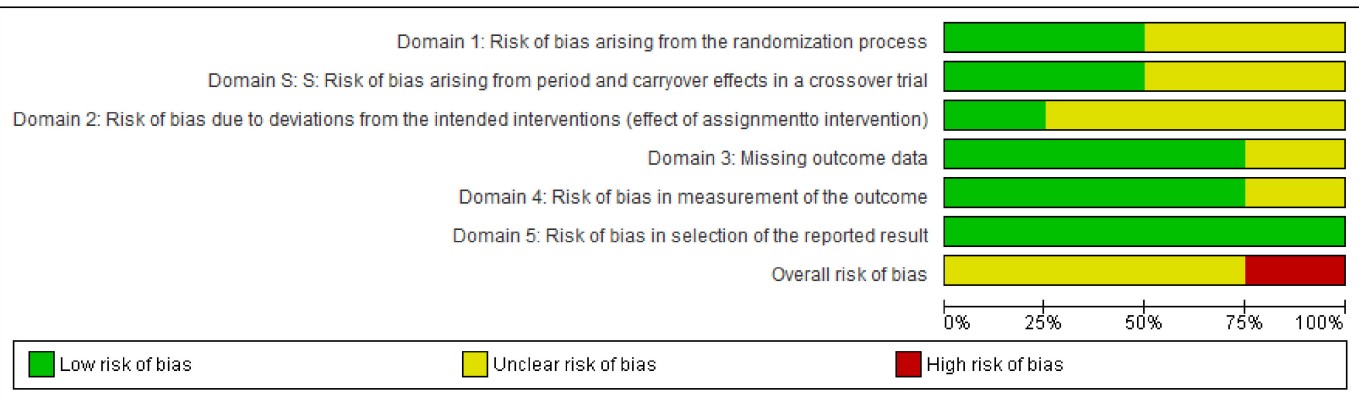

**Figure 3.** Authors' judgments about each item.

*3.3. Main Results*

3.3.1. Rapid Insulin-and-Pramlintide (CC) vs. Rapid Insulin-Alone (CC) vs. Regular Insulin-and-Pramlintide (CC)

Effectiveness

- **Clinical results.** In Haidar et al. (2020) [8], the dual system with rapid insulin was better than rapid insulin alone for time in target range (84% vs. 74%, $p = 0.001$), time > 180 mg/dL (12% vs. 22%, $p < 0.001$), average glucose (133 mg/dL vs. 144 mg/dL $p = 0.001$) and coefficient of variation of the glucose level (26.8% vs. 30.3%, $p = 0.035$). The results were only significant during the daytime period; at night, they were only significantly favorable to the intervention in terms of the standard deviation of glucose levels (36 mg/dL vs. 45 mg/dL $p = 0.002$). There were no significant differences in time in hypoglycemia or the number of insulin boluses. In the overnight period, the dual system with regular insulin was significantly worse than rapid insulin alone in time in target range (83% vs. 94%, $p = 0.002$), time < 70 mg/dL (<1% vs. <1%, $p = 0.006$) and >180 mg/dL (<1% vs. <1%, $p = 0.013$), and also for the coefficient of variation of the glucose level (24.6% vs. 15.9%, $p = 0.001$). During the 24 h, the dual system required more insulin boluses (25.5 vs. 22.6 units, $p = 0.002$) and basal insulin (27.5 vs. 23.8 units, $p = 0.048$).

Table 3 shows the effectiveness of the included studies.

Safety

With the dual system with rapid insulin and with insulin alone, the rate of patients with at least one hypoglycemic event (<60 mg/dL) that required treatment was similar (30% vs. 32%, respectively; no statistical contrasts were performed), as well as the number of events (12 vs. 11). In the dual system with regular insulin, the results were poorer (50% of patients had 18 events). During the phase of rapid insulin alone, no patient suffered gastrointestinal adverse events, but with the dual system with rapid insulin, there were 7% mild and 11% moderate events. In the dual system phase with regular insulin, 4% had mild gastrointestinal symptoms, 12% had moderate symptoms, and 8% had moderate-severe symptoms.

**Table 3.** Effectiveness results among the included studies.

| | Time in Target Range (70–180 mg/dL) | Time in Hypoglycemia | Time in Hyperglycemia | Mean Glucose Level (mg/dL) | Glucose Variability | Insulin Units |
|---|---|---|---|---|---|---|
| Haidar, (2020) [8] | RAI + PR (CC) vs. RAI (CC) 84% vs. 74% *p* = 0.001 <br><br> REGI + PR (CC) vs. RAI (CC) 69% vs. 74% *p* = 0.22 | RAI + PR (CC) vs. RAI (CC) **<70 mg/dL** 0% vs. 1.2% *p* = 0.43 **<60 mg/dL** 0% vs. 0% *p* = 0.78 <br> REGI + PR (CC) vs. RAI (CC) **<70 mg/dL** 7.3% vs. 1.2% *p* = 0.008 **<60 mg/dL** 1.2% vs. 0% *p* = 0.027 | RAI + PR (CC) vs. RAI (CC) **>180 mg/dL** 12% vs. 22% *p* <0.001 **>250 mg/dL** 0% vs. 0% *p* = 0.002 <br> REGI + PR (CC) vs. RAI (CC) **>180 mg/dL** 24% vs. 22% *p* = 0.49 **>250 mg/dL** 1% vs. 0% *p* = 0.37 | RAI + PR (CC) vs. RAI (CC) 133 vs. 144 *p* = 0.001 <br><br> REGI + PR (CC) vs. RAI (CC) 144 vs. 144 *p* = 0.95 | RAI + PR (CC) vs. RAI (CC) **SD (mg/dL)** 36 vs. 45 *p* = 0.002 **VC%** 25.6 vs. 29.3 *p* = 0.017 <br> REGI + PR (CC) vs. RAI (CC) **(mg/dL)** 45 vs. 45 *p* = 0.81 **VC%** 28.7 vs. 29.3 *p* = 1.00 | RAI + PR (CC) vs. RAI (CC) **Total basal** 15.3 vs. 15.3 *p* = 0.79 **Total boluses** 22.6 vs. 23.1 *p* = 0.35 <br> REGI + PR (CC) vs. RAI (CC) **Total basal** 19.4 vs. 15.3 *p* = 0.016 **Total boluses** 25.5 vs. 23.1 *p* = 0.002 |
| Tsoukas, (2021) [24] | FiASP + PR (SMA) vs. FiASP (CC) **Feasibility study** 84% vs. 81% **Pilot study** 70% vs. 70% | FiASP + PR (SMA) vs. FiASP (CC) **<70 mg/dL** **Feasibility study** 2.1% vs. 4.1% **Pilot study** 1.4% vs. 1.1% | FiASP + PR (SMA) vs. FiASP (CC) **>180 mg/dL** **Feasibility study** 14% vs. 13% **Pilot study** 28% vs. 28% | FiASP + PR (SMA) vs. FiASP (CC) **Feasibility study** 135 vs. 131 **Pilot study** 158 vs. 157 | FiASP + PR (SMA) vs. FiASP (CC) **SD (mg/dL)** **Feasibility study** 43 vs. 40 **Pilot study** 58 vs. 54 **VC%** **Feasibility study** 31.1 vs. 29.7 **Pilot study** 35.5 vs. 33.9 | FiASP + PR (SMA) vs. FiASP (CC) **Total basal** **Feasibility study** 28.7 vs. 27.4 **Pilot study** 33.0 vs. 30.9 **Total boluses** **Feasibility study** 8.4 vs. 18.1 **Pilot study** 13.0 vs. 18.4 |

**Table 3.** *Cont.*

| | Time in Target Range (70–180 mg/dL) | Time in Hypoglycemia | Time in Hyperglycemia | Mean Glucose Level (mg/dL) | Glucose Variability | Insulin Units |
|---|---|---|---|---|---|---|
| Tsoukas, (2021b) [25] | FiASP + PR (CLS) vs. FiASP (CC) 74.3% vs. 78.1% *p* = 0.28 (Non-inferiority contrast) | FiASP + PR (CLS) vs. FiASP (CC) **<70 mg/dL** 0% vs. 1.8% *p* = 0.058 **>60 mg/dL** 0% vs. 0% *p* = 0.32 | FiASP + PR (CLS) vs. FiASP (CC) **>180 mg/dL** 24.3% vs. 19.8% *p* = 0.093 **>234 mg/dL** 2.4% vs. 0.4% *p* = 0.56 | FiASP + PR (CLS) vs. FiASP (CC) 148 vs. 142 *p* = 0.060 | FiASP + PR (CLS) vs. FiASP (CC) **SD (mg/dL)** 49 vs. 45 *p* = 0.44 **VC%** 31.4 vs. 33.1 *p* = 0.95 | FiASP + PR (CLS) vs. FiASP (CC) **Total basal** 30.5 vs. 30.7 *p* = 0.42 **Total boluses** 18.8 vs. 27.9 *p* = 0.010 |

Note: The results of the FiASP (SMA) phase in Tsoukas et al. (2021), which had worse results than the two reported interventions, are not included. SMA: simple meal announcement; CC: carbohydrates counting; VC: variation coefficient; SD: standard deviation; FiASP: faster-acting insulin aspart (Ultra-rapid insulin); RAI: rapid insulin; REGI: regular insulin; PL: placebo; PR: pramlintide; CLS: close-loop system.

3.3.2. FiASP-and-Pramlintide System with SMA vs. FiASP System with CC vs. FiASP System with SMA

Effectiveness

- **Clinical results.** The crossover feasibility trial (n = 7) and the pilot study (n = 4) by Tsoukas et al. (2021) [24] did not perform statistical contrasts, given the small sample sizes. According to the feasibility study, the dual system with SMA had a shorter time under 70 mg/dL than the FiASP with CC (2.1% vs. 4.1%), a slightly longer time in target range (84% vs. 81%) and less use of boluses (8.4 vs. 18.1). In the pilot study, the time in hypoglycemia was longer with the dual system (1.4% vs. 1.0%), which also used fewer boluses (13.0 vs. 18.4). This study also included a phase of FiASP alone with SMA, which obtained clearly worse results than the other two interventions (results not shown in the table).
- **Quality of life.** In the pilot study, the dual system obtained lower diabetes-related stress scores (Diabetes Distress Scale, range 1–6) than FiASP alone (1.8 vs. 2.4) but slightly worse values in fear of hypoglycemia (1.6 vs. 1.4, Hypoglycemia Fear Survey-II, range 1–5).
- **Satisfaction with treatment.** The dual system scored slightly worse than FiASP alone (4.1 vs. 4.3, INSPIRE questionnaire, range 1–5).

Safety

During the FiASP-alone phase of the feasibility study, there were no complaints of gastrointestinal symptoms after any meal, whereas with the dual system, mild symptoms occurred in 12.9% of meals, moderate symptoms in 6.5%, and moderate-severe symptoms in 6.5% (none serious). In the pilot study, the phase with the dual system had slightly higher scores in frequencies (1.3 vs. 1.1) and severity (1.3 vs. 1.0) on the Diabetes Bowel Symptom Questionnaire (range 1–5). During the pramlintide phase, two patients showed skin irritation in the area of the infusion of this hormone, and another patient showed lipodystrophy in the area of the infusion of both hormones.

3.3.3. FiASP-and-Pramlintide System with no Meal Input (Fully Artificial Pancreas) vs. FiASP-Alone System with Precise CC (Hybrid Artificial Pancreas)

Effectiveness

- **Clinical results.** In the study by Tsoukas et al. (2021b) [25] (n = 28), the dual-release system was found to be non-inferior (within 6% of the time) to the hybrid system (without pramlintide) for time in target range (74.3% vs. 78.1%, $p = 0.28$). There were no significant differences in time in hypoglycemia and hyperglycemia or glycemic variability for the 24-h period in the superiority contrasts. For the AP system with pramlintide, results during the daytime (8:00–22:00) were significantly worse in terms of time in range (66.1% vs. 78.6%, $p = 0.016$), time above 180 mg/dL (32.7% vs. 20.8%, $p = 0.009$), and glucose level (160 vs. 150 mg/dL, $p = 0.018$) (data not shown in the table). The differences were not significant during the night. During the intervention phase, fewer insulin boluses were used (18.8 vs. 27.9, $p = 0.010$).

Safety

In the intervention phase, fewer participants experienced hypoglycemia (<60 mg/dL), although this difference was not significant (33% vs. 58%, $p = 0.15$), and similar results were found for hypoglycemia that required carbohydrate intake (11 vs. 21 events, no statistical comparison was made). There were more participants in the intervention phase who presented with some intestinal symptoms (29% versus 8%, no statistical comparison was made), one of them severe (nausea). There were no serious adverse effects or ketosis. One participant in the intervention phase showed skin irritation at the pramlintide insertion site. Table 4 shows the adverse effects observed among the included studies.

**Table 4.** Adverse events observed among the included studies.

| | Adverse Events |
|---|---|
| Haidar, (2020) [8] | RAI: 11 hypoglycemic episodes (HE) requiring oral treatment (1 HE every 2.5 d); gastrointestinal symptoms (GIS) 0% after (0 of 112) meals). RAI + PR: 12 HE (1HE every 2.3 d); GIS: 6% (6 out of 108 meals); 3 mild and 3 moderate, all were transient. REGI + PR: 18 HE (1 HE every 1.4 d); GIS: 11% (11 of 104 meals); 2 mild, 6 moderate and 3 moderate to severe. There were no elevated ketones (>18 mg/dl) in any administration of the artificial pancreas; non-inferiority contrast |
| Tsoukas, (2021b) [25] | 8 participants (33%) had at least one hypoglycemia event (<60 mg/dL) with the closed artificial pancreas vs. 14 (58%) participants with the hybrid system; 3 participants (13%) reported non-mild nausea and 1 participant (4%) with non-mild bloating with the artificial pancreas closed; no participant presented these events in the hybrid system. |
| Tsoukas, (2021) [24] | Feasibility study: RAI + PR with SMA: 3 participants experienced gastrointestinal symptoms. Pilot study: Frequency and severity of gastrointestinal symptoms measured with a Likert scale from 1 to 6: FIASP + placebo (CC) Frequency: 1.1 ± 0.4 Severity: 1.0 ± 0.2 FIASP + PR (SMA): Frequency: 1.3 ± 0.7 Severity: 1.3 ± 0.6 1 participant reported moderate abdominal pain FIASP + placebo con (SMA): Frequency: 1.3 ± 0.5 Severity: 1.3 ± 0.6 1 participant reported moderate abdominal pain and moderate nausea. |

HE: hypoglycemic episodes; GIS: gastrointestinal symptoms; CC: carbohydrate counting; SMA: simple meal announcement; PR: pramlintide; d: days; FiASP: faster-acting insulin aspart.

### 3.3.4. Ongoing Trials

We found two ongoing small crossover trials carried out by the same research group, with estimated completion dates of August (NCT05199714) and November 2022 (NCT04243629). They are described in Supplementary Table S3.

## 4. Discussion

People with T1DM require lifelong glucose monitoring and insulin replacement therapy, and several technological devices have been developed over the last decades to help them in the self-management of this disease, such as insulin pumps or continuous glucose monitoring devices [1,12]. The AP is the latest evolution of these technologies, with the main aim of enabling a fully closed-loop system to provide a completely automatized insulin administration, mimicking the biological pancreas [12]. However, current closed-loop systems still require that users enter the carbohydrate content of upcoming meals to determine the adequate dosing of pre-prandial insulin [6]. The incorporation of pramlintide within the AP system could help to delay mealtime insulin requirements by delaying gastric emptying, suppressing nutrient-stimulated glucagon secretion and increasing [18]. This may be of particular interest to those patients who do not achieve the expected clinical results with conventional insulin therapy.

The results of this review show that the scientific evidence on the efficacy and safety of this device is still very scarce, consisting of four small studies carried out in a 24-h inpatient setting (except one of them, with an outpatient follow up of 12 days but only 4 patients) by the same research group. Haidar et al. (2020) [8] found that the dual system

outperformed the insulin-alone system when both were applied along CC, while the results of Tsoukas et al. (2021b) [25] showed that the fully closed-loop dual system (without CC or SMA), although showed non-inferiority in time in therapeutic glucose range compared to the insulin-alone system with CC, obtained significantly worse outcomes during the daytime (relative increase of 57% of time above 180 mg/dL). When the dual system was accompanied by SMA in the two studies of Tsoukas et al. (2021) [24], results were very similar to that obtained by the insulin-alone system with CC.

Regarding safety, the three studies showed gastrointestinal adverse effects associated with the dual system, although they were non-serious in all cases. Previous studies about pramlintide administration have shown that nausea, the most common of these effects, is transient and dissipates within days to weeks.

In summary, the identified evidence is very limited, carried out by the same research group, with a total of 59 patients studied in a 24-h inpatient intervention (except for four patients). The review highlights the still-emerging nature of this technology and the need for additional technological research and development before conducting more externally valid outpatient studies, including pump chamber redesign and improvements in meal detection algorithms. Future studies should compare different pramlintide-to-insulin ratios, as well as develop optimal co-formulations of insulin and pramlintide, allowing the infusion of both hormones through the same conventional single-chamber insulin pump. Efficacy trials with longer follow-ups in outpatient settings are warranted, as well as the evaluation of the safety profile of the device. While the aim of AP is to improve glycemic control and reduce the need for decision-making regarding insulin infusion, the results show that control of postprandial hyperglycemia remains a challenge.

## 5. Conclusions

The current evidence on the effectiveness of a dual-hormone AP with insulin and pramlintide is very scarce and preliminary. The results of 24-h inpatient interventions suggest that it could outperform the single-hormone system when it is accompanied by an accurate CC before meals, but it could increase hyperglycemia during the daytime when no CC or SMA is made. Further, it could increase the occurrence of gastrointestinal adverse effects. Research in a real-world setting with longer follow-ups is warranted.

**Supplementary Materials:** The following supporting information can be downloaded at: https://www.mdpi.com/article/10.3390/app122010262/s1, Table S1: PRISMA check-list, Table S2: Search strategy, Table S3: Ongoing studies, Table S4: Risk of bias of crossover RCTs.

**Author Contributions:** Conceptualization, supervision, writing: A.T.-C.; original draft preparation, —reviewing and editing, A.D.-D.; formal analysis, data curation: A.R.-S.; conceptualization, writing—reviewing and editing: L.P.-P., A.A.-S., V.R.-G., Y.Á.-P.; conceptualization, writing—reviewing and editing: A.M.W., M.R., P.S.-A. All authors have read and agreed to the published version of the manuscript.

**Funding:** This work was financed by the Spanish Ministry of Health in the framework of activities developed by the Spanish Network of Agencies for Health Technology Assessment for the National Health Service.

**Institutional Review Board Statement:** Not applicable.

**Informed Consent Statement:** Not applicable.

**Acknowledgments:** The authors would like to thank Carlos Gonzalez Rodriguez and Leticia Rodriguez Rodriguez for their support as documentalists.

**Conflicts of Interest:** The authors declare no conflict of interest.

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
