# Peer review of "Dual-Hormone Insulin-and-Pramlintide Artificial Pancreas for Type 1 Diabetes: A Systematic Review"

_applsci, doi:10.3390/app122010262_

Round 1

Reviewer 1 Report (Previous Reviewer 2)

The authors attended to all the suggestions I made in the manuscript. This systematic review is well structured, and I recommend the acceptance for publication is in this improved version.

Author Response

Reviewer 1

The authors attended to all the suggestions I made in the manuscript. This systematic review is well structured, and I recommend the acceptance for publication is in this improved version.

Thanks for your comments.

Reviewer 2 Report (New Reviewer)

If I understand correctly, this systematic review was less about the use of the artificial pancreas but more specific to the infusion of Pramlintide using an AP? I would make this clearer in the paper. Just this year a meta-analysis on this very topic was published by Zeng et al 2022: https://dom-pubs.onlinelibrary.wiley.com/doi/10.1111/dom.14781 which came across 17 trials equating to 438 patients. All but one (Haidar 2020) identified were relating to insulin & glucagon APs. So maybe it is worth spelling out that the infusion of glucagon is more common? And why not compare this with the pramlintide infusions? Given Pramlintide is so new, why do a systematic review at this time? It's no surprise that there are few studies with very small numbers which provide inconclusive results other than we need more research on this hormone infusion using AP?

Four small studies (two of the pilots) from the same researchers in the same population provides very little generalisability. This systematic review is based on a total of 59 patients and might be worth emphasising these small numbers in the paper more. Thus your conclusion that it performs worse on hyperglycaemia in daytime based on one of the small studies is a stretch at best. What does this paper add? That it is specific to pramlintide infusion using AP, and I think this needs to be fleshed out more. Not making conclusions of AP generally as the abstract suggests.

Line 106 mentions Clinicaltrials.gov was also used in your search strategy. I would include in the abstract. I can't seem to find the supplementary materials, but did you find additional papers from your original 4 from searching this database? I'm not quite sure why you included this here otherwise.

Strongly suggest that abbreviation T1D or T1DM be used over DM1. Please change throughout the text. If you conduct a search on DMQ you're more likely to retrieve myotonic dystrophy 1. The common abbreviation for type 1 diabetes is T1D. 

Line 49 GMS is used as an abbreviation not CGM. Then CGM is used from then on. Please correct.

In the search strategy did you type in "type 1 diabetes"? It just seems glaringly omitted. 

Line 276 "non-inferior" seems such a strange term. Why not equivalent or similar....?

Line 310 typo "veryscarce"

Line 326 and great number of patients from a wider population!

Line 336 typo "will easy adoption."

Line 354 "follow up are warranted." suggest change to "is warranted" as research is an example of a mass noun, and you should always write it in the singular form.  

Author Response

Reviewer 2

If I understand correctly, this systematic review was less about the use of the artificial pancreas but more specific to the infusion of Pramlintide using an AP? I would make this clearer in the paper. Just this year a meta-analysis on this very topic was published by Zeng et al 2022: https://dom-pubs.onlinelibrary.wiley.com/doi/10.1111/dom.14781 which came across 17 trials equating to 438 patients. All but one (Haidar 2020) identified were relating to insulin & glucagon APs. So maybe it is worth spelling out that the infusion of glucagon is more common? And why not compare this with the pramlintide infusions? Given Pramlintide is so new, why do a systematic review at this time? It's no surprise that there are few studies with very small numbers which provide inconclusive results other than we need more research on this hormone infusion using AP?

Yes, the paper effectively focuses on the effect of adding pramlintide to a single-hormone AP. The use of pramlintide is aimed to avoid postprandial hyperglycemia, and ultimately, to eliminate the need of carbohydrate counting before meals. It is not a competing alternative to the use of glucagon, which is aimed to avoid hypoglycemia induced by external insulin. Therefore, dual-hormone systems with glucagon infusion are not relevant for this review. We have made changes in the abstract and the introduction to clarify this.

            We knew that AP with pramlintide is a recent development, but before carrying out the review we could not know if there was at least one trial of good quality. Publishing a systematic review with little evidence is an editorial decision, and it is a fact that even “empty reviews” are published [1].

[1] Yaffe J, Montgomery P, Hopewell S, Shepard LD. Empty reviews: a description and consideration of Cochrane systematic reviews with no included studies. PLoS One. 2012;7(5):e36626.

Four small studies (two of the pilots) from the same researchers in the same population provides very little generalizability. This systematic review is based on a total of 59 patients and might be worth emphasising these small numbers in the paper more. Thus your conclusion that it performs worse on hyperglycaemia in daytime based on one of the small studies is a stretch at best. What does this paper add? That it is specific to pramlintide infusion using AP, and I think this needs to be fleshed out more. Not making conclusions of AP generally as the abstract suggests.

The paper describes the current evidence on the topic. It concludes that it is very scarce and limited, but we think that this conclusion is not equivalent to a complete absence of studies. The results show that the use of pramlintide could outperform the insulin-alone AP when a precise carbohydrate counting is maintained (Haidar 2020), but it may function significantly worse without it (Tsoukas 2021), and that gastrointestinal effects were consistently higher with pramlintide.

            If we assume that this knowledge is equivalent to no knowledge and therefore not worth publishing, this would be all the more applicable to any small or pilot study such as those included in this review.

            Nonetheless, we have made changes throughout the text to emphasise the weakness of the evidence.

Line 106 mentions Clinicaltrials.gov was also used in your search strategy. I would include in the abstract. I can't seem to find the supplementary materials, but did you find additional papers from your original 4 from searching this database? I'm not quite sure why you included this here otherwise.

No, we only found two ongoing trials from the same research group (NCT04243629; NCT05199714). We have included in the abstract that we also searched the ClinicalTrials.gov database, and we have described them in the supplementary file 3. All supplementary files have been successfully uploaded to the journal's electronic platform.

Strongly suggest that abbreviation T1D or T1DM be used over DM1. Please change throughout the text. If you conduct a search on DMQ you're more likely to retrieve myotonic dystrophy 1. The common abbreviation for type 1 diabetes is T1D. 

Done.

Line 49 GMS is used as an abbreviation not CGM. Then CGM is used from then on. Please correct

Done.

In the search strategy did you type in "type 1 diabetes"? It just seems glaringly omitted. 

Diabetes Mellitus, Type 1, was included in the search strategy. As can be seen in Table 1, Medline search strategy in line 1.

Line 276 "non-inferior" seems such a strange term. Why not equivalent or similar....?

The term equivalent has been incorporated rather than not inferior.

Line 310 typo "veryscarce"

It was corrected.

Line 326 and great number of patients from a wider population!

Corrected in the previous revision.

Line 336 typo "will easy adoption."

It was corrected.

Line 354 "follow up are warranted." suggest change to "is warranted" as research is an example of a mass noun, and you should always write it in the singular form.  

It was corrected.

Reviewer 3 Report (New Reviewer)

1. Authors mentioned in the abstract and methods that databases were consulted from inception. Authors need to specify the date of inception.

2. Please arrange keywords alphabetically.

3. Introduction was not well referenced.

4. Discussion was not done with enough references.

5. List of references is too scanty for a review. 

Author Response

Reviewer 3

  1. Authors mentioned in the abstract and methods that databases were consulted from inception. Authors need to specify the date of inception.

The date of the search has been corrected.

  1. Please arrange keywords alphabetically.

It was corrected

  1. Introduction was not well referenced.

We have made changes in the text based on the comments of another reviewer, and 12 new references have been included.

  1. Discussion was not done with enough references.

We have made changes in the text based on the comments of another reviewer, and 12 new references have been included.

  1. List of references is too scanty for a review. 

New references have been added. In any case, we would like to emphasize that our interest is not to document the artificial pancreas, but specifically the one that has a dual release of insulin and pramlintide, of which there are few published studies.

This manuscript is a resubmission of an earlier submission. The following is a list of the peer review reports and author responses from that submission.

Round 1

Reviewer 1 Report

This paper review the current state of the dual-hormone insulin-pramlintide artificial pancreas (Dual-AP) applied to subjects with type 1 diabetes, comparing its performance, in terms of glucose control efficacy and safety, against single hormone (insulin) AP. Specifically, it is stated the non-inferiority of Dual-AP vs. AP in the 24 hours, while its performance worsen when focusing on daytime or overnight. Nevertheless, the lower performance is somewhat justified by the young-age technology, so that more solid comparison will be possible in the future.

My comments are reported below

1) Despite the technology subject of the review is very recent, the reference list must be improved. For instance, a brief historical overview of the AP technology, as well as its state of the art, should be properly discussed, with particular focus to single-hormone AP algorithms (hybrid, fully-automated, adaptive) and dual insulin-glucagon ones. This also will contribute to better justify the use of dual insulin-pramlintide AP.
Moreover, I am wandering if a review on this topic is appropriate at this early stage of development.
2) On the same aspect, it should be better discussed why the dual insulin-pramlintide AP is promising, why it should be preferred, and for which possible target population.
3) It is not clear if all the comparisons refer to cross-over trials. If not, baseline characteristics and glucose levels must be provided to show if they are well matched among the different cohorts, otherwise, it might affect the statistical comparison.
4) The hardware setup employed in the different AP must be described in terms of glucose sensor, hormone delivery device, closed-loop control algorithm.
5) In the results and conclusion sections, the authors discuss about non inferiority of dual- vs. single-hormone AP. However, this is a delicate statement that must be properly discussed and demonstrated from a statistical point of view. Therefore, I suggest the authors to change the terminology, unless they are able to provide statistical power analysis with minimum effect size.
6) Minor issue: at line 218, time above target should be >180 mg/dl.

Reviewer 2 Report

The manuscript shows a detailed systematic review previously registered in PROSPERA, but the number of authors is different.

Inconsistencies are shown in the use of units throughout the text that must be unified and some terms and spaces between words that must be corrected.

The quality of figure 1 should be improved.

The author cites reference number 11, and the number of references is until 10.

The manuscript contains relevant information and is, in general, well presented.

Reviewer 3 Report

This manuscript is devoted to a very relevant and socially significant topic – the treatment of type 1 diabetes mellitus through the use of an artificial pancreas with double hormonal insulin and pramlintide.

In lines 74-177, the technology of the review was described, according to the results of which only 4 publications of scientists from one research group from Canada (Tsoukas, Haidar, Rutkowski et al.) were actually selected. No other publications were found by the authors. There are only 10 sources of information in the bibliographic list of the article.

What is the scientific value of this review, if in fact the only conclusion is "the scientific evidence on the effectiveness of this dual AP system is still urgent and very scarce, consisting of four small studies (described in three articles) carried out in an appropriate setting by the same research group" (lines 298-300)?

From the analysis it turns out that this AP has not yet been investigated and there are few published results? Why? The manuscript doesn't say anything about it.

Perhaps the search criteria need to be expanded by adjusting paragraphs (3) and (4) (lines 109-111).

In any case, I consider it inappropriate to publish "a systematic review" based on the results of a single scientific group.

This publication gives nothing to scientists and doctors, while millions of patients with type I diabetes and their relatives are waiting for the development of an effective treatment or at least compensation for diabetes.

Round 2

Reviewer 1 Report

I thank the authors to have tried to address my critics.

However, I believe that the main issue remains. That is I still am not convinced about a systematic review paper at this too preliminary stage.

I’m not saying that the topic is not interesting. Nevertheless, a review on such a new topic should analyze more contributions, coming from different research groups, in order to provide more solid contribution.

Reviewer 3 Report

I still consider it inappropriate to publish the manuscript as a Review. I suggest that the authors write a Research Article, taking part of this manuscript as an Introduction and supplementing it with their own studies of an artificial pancreas with double hormonal insulin and pramlintide.